# Characterization of Normal Bone in the Equine Distal Limb with Effective Atomic Number and Electron Density Determined with Single-Source Dual Energy and Detector-Based Spectral Computed Tomography

**DOI:** 10.3390/ani14071064

**Published:** 2024-03-30

**Authors:** Janine Steiner, Henning Richter, Rolf Kaufmann, Stefanie Ohlerth

**Affiliations:** 1Clinic of Diagnostic Imaging, Vetsuisse Faculty, University of Zurich, 8057 Zurich, Switzerland; janinecarmen.steiner@uzh.ch (J.S.); henning.richter@uzh.ch (H.R.); 2Effectum Medical AG, 4600 Olten, Switzerland; rolf@kaufmannweb.ch

**Keywords:** computed tomography, dual energy, effective atomic number, electron density, equids, subchondral bone

## Abstract

**Simple Summary:**

Diseases involving the subchondral bone are common in horses and are usually evaluated with conventional imaging techniques, such as radiography, conventional computer tomography, and magnetic resonance imaging. Characterization of the normal subchondral bone as well as exercise-related changes is of special interest in ongoing research. Single-source dual energy and detector-based spectral computed tomography are emerging technologies allowing the characterization of tissues that have different attenuation properties at different energies. Both technologies enable the calculation of effective atomic number, an index to determine tissue composition, and electron density, which is assumed to be associated with cellularity in tissues. In the present study, both values were determined in normal subchondral and trabecular bones in the distal limb of horses with single-source dual energy and detector-based spectral computed tomography. Several effective atomic number and electron density values differed between both technologies and within and between bones of the distal equine limb with both technologies. Effective atomic number values increased with age in the palmar/plantar fetlock joint zone. The reported effective atomic number and electron density values in the subchondral and trabecular bone of the equine distal limb may serve as preliminary reference values and aid future classification of diseases.

**Abstract:**

Single-source dual energy (SSDECT) and detector-based spectral computed tomography (DBSCT) are emerging technologies allowing the interrogation of materials that have different attenuation properties at different energies. Both technologies enable the calculation of effective atomic number (EAN), an index to determine tissue composition, and electron density (ED), which is assumed to be associated with cellularity in tissues. In the present prospective observational study, EAN and ED values were determined for 16 zones in normal subchondral and trabecular bone of 37 equine cadaver limbs. Using both technologies, the following findings were obtained: 1. palmar/plantar EAN zone values in the fetlock increased significantly with increasing age of the horse; 2. all EAN and ED values were significantly lower in the trabecular bone than in the subchondral bone of all phalanges; 3. in the distal phalanx and navicular bone, most EAN and ED values were significantly lower compared to the proximal and middle phalanx; and 4. some EAN and ED values were significantly different between front and hind limbs. Several EAN and ED values significantly differed between SSDECT and DBSCT. The reported EAN and ED values in the subchondral and trabecular bone of the equine distal limb may serve as preliminary reference values and aid future evaluation and classification of diseases.

## 1. Introduction

Dual energy computed tomography (DECT) represents an emerging and promising imaging technology in the medical field. Whereas conventional CT generally uses a single polychromatic photon spectrum, DECT uses two polychromatic photon spectra with different maximum energies, allowing the interrogation of materials/tissues that have different attenuation properties at these particular energies [1]. Several DECT technologies are currently available. Single-source DECT (SSDECT) with rapid voltage switching uses one tube, which rapidly changes between low- and high-energy beams. The two projection data sets are collected separately for subsequent use in a projection-based dual-energy reconstruction algorithm. With detector-based spectral CT (DBSCT), a single tube produces a polychromatic photon beam at a single fixed tube voltage as in conventional CT. A dual-layer detector then absorbs selectively low-energy photons at the inner layer, whereas the outer layer absorbs high-energy photons [2].

In conventional CT, attenuation of the photon beam by a material is related to the linear attenuation coefficient, which is not unique to any given material as the same linear attenuation coefficient can be measured for two different materials at a given photon energy. The CT number (Hounsfield units) measures the linear attenuation coefficient and generally depends not only on the material’s atomic number but also on its density and photon energy. It follows that two different materials with different elemental compositions can have the same CT number [3].

Whereas most medical applications of DECT describe the use of qualitative evaluation of tissues, DECT also enables quantitative assessment of materials. For example, effective atomic number (EAN) and electron density (ED) may be calculated from DECT data with small errors of 1.7 and 4.1%, respectively [4]. In the stricter sense, the unitless EAN describes the effective nuclear charge of an atom; however, it also calculates the average atomic number for a compound or mixture of materials based on the idea that the EAN assumes that a mixture or a compound can be regarded for special purposes as being comprised of one kind of particle/atom with the atomic number Z [5]. Electron density is the measure of probability of an electron being present in a specific location and is expressed as a multiple of 10^23^ electrons/mL [3]. Increasing cellularity with increasing tumor grade translated into increasing ED as cells are closely packed together with a high mitotic index and nucleo-cytoplasmic ratio indicates a higher number of electrons per unit volume of tissue [3,6].

Since compounds of different chemical compositions may have similar linear attenuation coefficients but different EAN or ED values, the latter may be used for further compound characterization of materials or tissues [7]. Whereas the most recent literature in human medicine reports on the qualitative use of DE, the application of EAN and ED measures has less commonly been investigated [8,9,10,11]. Electron density measures play a key role in radiation therapy [12]; however, the diagnostic use has only been described recently. For example, DECT-based ED measurements were shown to be helpful for the differentiation of gliomas [3], the prediction of vascular density in meningiomas [13], and the diagnosis of metastatic lymph nodes in non-small cell lung cancer [14], pulmonary embolism [15], and spinal hematoma [16]. In forensic medicine, a prediction algorithm based on EAN and ED values was successful in distinguishing a range of explosives in a human body phantom [17].

In the human musculoskeletal system, reports on the use of EAN and ED maps are sparse. Electron density images improved cervical disc herniation detection and diagnostic confidence of CT [18]. In the human knee, EAN and ED values significantly differed between calcium pyrophosphate depositions and calcium hydroxyapatite in subchondral and trabecular bone [19]. So far, there is only limited information on the applicability, possible use, and pitfalls of DECT in veterinary and especially equine medicine [20,21,22]. Recently, increased attenuation was described subjectively with SSDECT on virtual non-calcium maps in two equine cadaver feet with bone edema-like lesions [23]. However, the authors stated that effects of age- or breed-dependent or pathologically increased bone density may play a significant role and need to be further addressed in future studies. Quantitative analyses, such as EAN or ED measurements, may aid in answering these questions. To the best of our knowledge, the use of EAN or ED maps in equine bone has not been described so far.

In both the equine as an athlete or a recreational partner, injuries and diseases involving the subchondral bone are common and are evaluated with conventional imaging techniques, such as radiography, conventional CT, and magnetic resonance imaging (MRI) [20,21,22]. The subchondral bone is basically composed of the subchondral bone plate and the adjacent trabecular bone. The term subchondral bone disease includes a wide spectrum and different stages of subchondral bone pathologies [24]. Subchondral bone disease is particularly common in racehorses; however, it is also well-known in many other types of horses and disciplines. The characterization of the subchondral and trabecular bone as well as exercise-related changes within bones of the distal limb, in particular, the distal metacarpus and metatarsus and the proximal phalanx, is of special interest in ongoing research [25,26,27,28,29]. Therefore, the purpose of the present study was to determine EAN and ED values in the normal subchondral and trabecular bone in the distal limb of horses calculated with SSDECT and DBSCT.

## 2. Materials and Methods

### 2.1. Specimen Collection

Cadaveric legs were harvested from April 2022 to February 2023. Horses presented at the Equine Clinic, Vetsuisse Faculty of Zurich, for reasons unrelated to lameness on the harvested distal legs and were finally slaughtered or euthanized. Owners’ consent was obtained for the present study. Within 2 h after death, legs were disarticulated proximal to the carpometacarpal or tarsometatarsal joint, and shoes were removed.

### 2.2. Diagnostic Imaging

After disarticulation, all legs underwent radiography first. Then CT and MRI were performed in 34 limbs within 8 h after death. In 3 subjects, MRI was performed within 72 h after death and in between, the legs were stored at 4 °C. Prior to MRI, they were kept at room temperature for 2 h.

Radiography—A 60° dorsopalmar view of the hoof was obtained to check for metallic particles, which were then removed. A dorsopalmar and lateromedial overview of the distal limb centered on the fetlock joint were also taken.

Computed tomography—In all limbs, the region from the distal third of the third metacarpal/metatarsal bone to the distal phalanx was scanned twice from proximal to distal with the palmar/plantar surface to the tabletop using two different CT units. For SSDECT, scans were performed using the volume scan mode and a rotate-rotate technique (CANON Aquilion Exceed LB, Canon Medical Systems, Wallisellen, Switzerland). Two sequential volume scans were acquired with a z-coverage of 4 cm using 80 kVp/460 mAs and 135 kVp/80 mAs, separately. Multiple volume scans were performed consecutively to enable a larger z-coverage. The following settings were used: rotation time 0.5 s, switching time 0.4 s, slice thickness 0.5 mm, slice interval 0.25 mm, scan field of view (S-FOV) 320 mm, and a display field of view depending on the sample size (D-FOV = 166.25–247.5 mm). Blended images representing a 120 kVp scan were calculated in a bone and a soft tissue kernel based on the DE images using proprietary software. For DBSCT, a dual-layer CT scanner was used (IQon Spectral CT, Philips Healthcare, Horgen, Switzerland). Images were obtained using a helical scan mode with a tube voltage of 120 kVp with automated attenuation-based tube modulation with the following settings: rotation time 0.75 s, pitch 0.671, slice thickness 1 mm, and a variable FOV (S-FOV = 143–198 mm) depending on the sample size. For reconstruction of conventional and spectral-based images, IDose level 0 and spectral level 2 were used, respectively (Philips Healthcare).

### 2.3. Magnetic Resonance Imaging

A 3 Tesla scanner and a dStream HeadNeckSpine coil were used (Philips Ingenia, Philips AG). Three sequences were acquired: a transverse T2-weighted turbo spin echo (T2w TSE) sequence from the proximal margin of the proximal sesamoid bones to the insertion of the straight sesamoidean ligament and a T1-weighted 3D (T1w3D, TFE SENSE) and a dorsal fat-saturated STIR sequence from the mid-third of the third metacarpal/metatarsal bone to the tip of the toe. The following parameters were used: T2wTSE: echo time (TE) 100 ms, repetition time (TR) 3660 ms, slice thickness 3 mm with an interslice gap of 3.3 mm, voxel size 0.6 × 0.7 × 3 mm, number of signal average (NSA) 1, bandwidth (BW) 146 Hz/pixel, echo train length 17, and FOV 150 × 169 × 115 mm; STIR: TR 7832 ms, TE 60 ms, IR 210, FOV 320 × 213 × 152 mm, TSE factor 11, matrix 456 × 220, slice thickness 3 mm, slice gap 3.3 mm, and NSA 2; T1w3D: TE 11.52, TR 5.23, slice thickness 0.6 mm without interslice gap, voxel size 0.6 × 0.6 × 0.7 mm, NSA 1, flip angle 8°, BW 134 Hz/pixel, echo train length 217, FOV 280 × 241 × 230 mm.

### 2.4. Image Analysis and Measurements

Radiographic, CT, and MRI images were transmitted to a diagnostic workstation and reviewed by a board-certified radiologist (SO) with dedicated software using multiplanar and 3D-reconstruction modes whenever applicable (IntelliSpace PACS Radiology, Version 4.4, Philips). Including all conventional imaging modalities, abnormalities were noted and scored on the level joint for the metacarpophalangeal or metatarsophalangeal joint (fetlock joint (FJ)) and the proximal (PIPJ) and distal (DIPJ) interphalangeal joint as follows: score 0 (normal), score 1 (minor findings of no clinical relevance and outside the CT region of interest (ROI)), and score 2 (major findings of considerable relevance and/or within a CT ROI). A joint was classified normal if the subchondral bone was uniform and of a thickness ≤3.5 mm and signs of adaptive or maladaptive changes or any other articular pathology were not present. For a score of 1, the following changes outside a CT ROI were included: focal subchondral bone thickness >3.5 mm, a non-uniform subchondral bone, or small subchondral bone lesions <3 mm (clefts or small cysts). All other maladaptive changes (osteophytes, cartilage lesions, subchondral bone lesions >3 mm outside a ROI or any subchondral bone lesion/sclerosis within a ROI) or any other articular pathology were assigned a score of 2. In the navicular bone (NB), a score of 1 included minor changes at its borders, e.g., irregularities or small fragments. A score of 2 was allocated to a NB if any kind of abnormality was seen in the spongious bone, such as widened synovial channels, cyst-like lesions, or sclerosis. Further, in the FJ, the location of subchondral bone lysis (clefts, cysts, patchy subchondral bone) and sclerosis was allocated to 16 anatomical regions: 7 regions for the distal MC/MT3 (dorsal, middle and palmar/plantar area of the lateral/medial condyle, and the sagittal ridge), 7 regions for the proximal P1 (dorsal, middle and palmar/plantar area of the lateral and medial fovea, and the sagittal groove), and one region per PSB.

Based on the CT raw data, EAN and ED maps were created from both SSDECT and SBCT images (slice thickness 2 mm), and ROIs were drawn in the virtual monochromatic images (window level 500, window width 2500) using the proprietary software (Raw Data Analysis, CANON Medical and Intellispace Portal 12, Philips Healthcare). Mean EAN/ED values were recorded for each ROI. In total, 29 ROIs and 16 zones were defined for each distal limb. A zone referred to a certain anatomical area, and the mean EAN/ED was calculated from all ROIs included in this area. In the distal aspect of the third metacarpal (MC) and metatarsal (MT) bone, a mid-dorsal image was chosen, and one ROI each was placed in the subchondral bone of the sagittal ridge and the medial and lateral condyle, respectively (dorsal MC/MT zone) (Figure 1). Additionally, a transverse image was chosen at the level of the maximum width of the proximal sesamoid bones (PSB), i.e., at the transition from the distal to the mid-third of the PSBs, and 1 ROI each was placed in the palmar/plantar subchondral bone of the medial and lateral metacarpal/metatarsal condyle (plantar MC/MT zone) and the dorsal subchondral bone of the medial and lateral proximal sesamoid bone (PSB zone) (Figure 2). The mean of all palmar/plantar ROIs in the FJ area (plantar MC/MT and PSB ROIs) represented the palmar/plantar FJ zone (Figure 2). In the first (P1) and second (P2) phalanx, a mid-dorsal image was chosen at the level of the marrow cavity or nutrient foramen, respectively, and in the third phalanx (P3), a dorsal image was chosen parallel to the dorsal surface of the bone at the level of the solar canal. Three ROIs each (medial and lateral fovea, sagittal groove or extensor process) were drawn in the proximal subchondral bone (proximal P1/P2/P3 subchondral zone) and trabecular bone (proximal P1/P2/P3 trabecular zone) distal to the latter in all phalanges (Figure 1 and Figure 3). Larger zones were calculated including all ROIs in the proximal subchondral and trabecular bone of P1, P2 and P3, respectively (proximal epiphyseal P1/P2/P3 zone) (Figure 1 and Figure 3). Moreover, the mean of all dorsal ROIs in the FJ area (distal MC/MT, proximal P1) represented the dorsal FJ zone (Figure 1). A single ROI was also applied to the marrow cavity of P1 (Figure 1). On a mid-dorsal plane of the NB, 3 ROIs were evenly distributed medially, centrally, and laterally within the spongious bone (navicular zone) (Figure 4). All ROIs were of a circular shape with a diameter of 4 mm (smallest ROI possible for both CT units) except for the circular ROI in the marrow cavity of P1, the diameter of which was adapted to the size of the marrow cavity.

### 2.5. Statistical Analysis

Data sheets evaluating the presence of radiographic, CT, and MRI findings were compiled for each limb. The history, presence of lameness (no; yes), age of the horse, and the extremity (right, left; front, hind) were also noted. Descriptive statistics were calculated for categorized and continuous variables. Due to a small sample size, nonparametric tests were applied. Statistical analyses were performed on the level of each joint (FJ, PIPJ, DIPJ) or NB. The Mann–Whitney U test was used for comparison of two independent samples, i.e., EAN/ED values between both CT units for left and right or front and hind legs and sound and lame horses, separately. The Kruskal–Wallis test was used for comparison of 3 independent samples, i.e., association of EAN/ED values with joint score and comparison of EAN values between the normal proximal P1, P2, and P3 zones. Associations between continuous variables (age, EAN/ED values) were analyzed with Spearman correlation coefficients. Associations between categorized variables (gender, breed) and continuous variables were not calculated because of uneven and small cell numbers. For testing the difference in median EAN/ED zone values between the subchondral and trabecular bone within P1, P2 or P3, separately, the Wilcoxon signed-rank test was applied. Due to a small sample size, multivariate testing was not applied.

Statistical analyses were performed using a commercially available software (SPSS statistics, release 27.0.0.0, IBM Corporation, Armonk, NY, USA). A *p*-value < 0.05 was considered significant.

## 3. Results

Animals—In total, 37 cadaveric limbs of 12 horses (10 warmbloods, 1 light draught type horse, 1 Spanish horse) were included, consisting of 18 right and 19 left legs as well as 18 front and 19 hind legs. All 111 synovial joints of the 37 specimens were included, of which 36 legs were scanned with both CT units. Due to technical problems, 1 limb each was only scanned with either SSDECT or SBCT; consequently, only 36 limbs (108 joints) were evaluated with each CT unit. Two mares and 10 geldings were represented. Median age was 15 years (SE, 1.5 years; range, 9–27 years). Reasons for euthanasia or slaughter included chronic lameness on a limb not included in the study (5 horses/13 limbs), acute thoracic or pelvic trauma (2 horses/6 limbs), and reduced performance, sinusitis, spermatic cord stump infection, retinal detachment, and unknown reason in one horse each (18 limbs).

Imaging Score—Overall, 7 (19%) legs were scored normal, 9 (24%) limbs had a maximum score of 1, and 21 (57%) subjects had a score of 2 in one or all joints. On the joint level, 61 (55%) joints were normal, and a score of 1 or 2 was allocated to 22 (20%) and 28 (25%) joints, respectively. The FJ was the most affected joint (Table 1); scores of 1 and 2 were observed in 44% and 43% with SSDECT (Canon) and DBSCT (Philips), respectively. Of all FJs that were scored 1 and 2, subchondral lysis (clefts, cysts, patchy subchondral bone) and subchondral bone thickness > 4 mm were by far less common in the PSB (8% and 12.5%) than in the MC/MT3 (37.5% and 32%) and P1 (70.1% and 92%). Of all MC/MT3s with subchondral lysis, the sagittal ridge, the dorsal area of the medial condyle (33.3% each), the medial area of the lateral and medial condyle as well as the palmar area of the medial condyle were affected (11.1% each). Subchondral bone thickness > 4 mm was observed in the dorsal area of the medial condyle (88%) and the lateral condyle (25%) and in the palmar area of the medial condyle and the middle area of the lateral condyle (12.5% each) of all affected MC/MT3s.

Of all P1s with subchondral lysis, the sagittal groove (70.6%), the middle area of the medial fovea (35%), and the lateral fovea (11.8%) were affected, whereas dorsal and plantar areas were not affected at all. Similarly, subchondral bone thickness > 4 mm was observed in the middle area of the medial fovea (70%), in the sagittal groove (61%), in the dorsal area of the medial fovea (52.2%), diffusely (26.1%), and in the middle and dorsal area of the lateral fovea (17.4% each) of all affected P1s. In the NB, a score of 0 was given to 32 (89%) specimens, and a score of 1 or 2 was allocated to 2 (5.5%) NBs each. The marrow cavity of P1 was scored normal in all specimens. Descriptive statistics of the normal EAN and ED zone values (score 0) are shown in Table 2 and Table 3.

*Comparison of normal EAN and ED zone values between CT units*—Median EAN values were significantly higher with SSDECT (Canon) in the proximal P1 trabecular zone (*p* = 0.018); the proximal P2 trabecular and epiphyseal P2 zone; the proximal P3 subchondral, trabecular, and epiphyseal zone; and the navicular zone (*p* < 0.001). In contrast, median ED values were significantly lower with SSDECT (Canon) for the proximal P1 epiphyseal zone; the proximal P2 trabecular and epiphyseal P2 zone; and the proximal P3 subchondral, trabecular, and epiphyseal zone (*p* < 0.001–0.017). Similarly, in the normal marrow cavity of P1, median EAN values were significantly higher and median ED values were significantly lower with SSDECT (Canon) compared to DBSCT (Philips) (*p* < 0.001). The remaining ED and EAN zone values did not significantly differ between the CT units (*p* = 0.157–1).

*Association of EAN/ED values with age of horses*—In normal fetlock joints, there was a strong positive correlation between age and two palmar/plantar EAN zone measurements: the PSB zone (for both CT units) and the palmar/plantar FJ zone (for DBSCT only) (*p* = 0.01–0.028, *r* = 0.68–0.76). In contrast, in normal PIPJs, there was a significant moderate negative correlation between age and EAN values of the proximal P2 subchondral zone for SSDECT (Canon) only (*p* = 0.016, *r* = −0.52). Age was significantly positively correlated with ED values of the normal marrow cavity of P1 with DBSCT (Philips) (*p* = 0.027, *r* = 0.36), and the correlation was borderline significant with SSDECT (Canon) (*p* = 0.05, *r* = 0.33). Age did not correlate with the remaining EAN and ED zone values (*p* = 0.05–0.966, *r* = −0.42–0.62) of both CT units.

*Comparison of front/hind normal EAN/ED values*—In the front PIPJ, median EAN and ED values of the proximal P2 trabecular zone were significantly higher for SSDECT (Canon) and DBSCT (Philips) (*p* = 0.003–0.013). Similarly, in the front DIPJ, median EAN values of the proximal P3 trabecular zone were significantly higher for SSDECT (Canon) (*p* = 0.009), but not for DBSCT (Philips) (*p* = 0.09). In the normal marrow cavity of P1, median ED values were significantly higher in hind legs with SSDECT (Canon) (*p* = 0.04). The remaining EAN and ED zone values were not significantly different (*p* = 0.059–0.71).

*Comparison of right/left normal EAN/ED values*—Median EAN and ED values of normal joints, NBs, and marrow cavity did not differ between left and right limbs for both CT units (*p* = 0.112–1.00).

*Comparison of EAN/ED values within/between normal bones*—With both units, EAN and ED values of all trabecular zones in the normal phalanges were lower than the values of the subchondral zones (*p* < 0.001–0.008). Further, EAN and ED zone values were compared between the normal proximal P1, P2, and P3. For SSDECT (Canon), the median EAN value of the proximal P3 subchondral zone was significantly lower compared to P2 and P1 (*p* = 0.001 and 0.023). Similarly, the median EAN value of the proximal P3 epiphyseal zone was significantly lower compared to P2 (*p* = 0.001) but not to P1 (*p* = 0.327). Regarding electron density, values of the proximal P3 subchondral and epiphyseal zone were significantly lower compared to P2 (*p* < 0.001 and 0.002) but not to P1 (*p* = 0.166 and 0.247). Median EAN and ED values of the proximal trabecular zone did not significantly differ between all phalanges (*p* = 0.088–1).

For DBSCT (Philips), median EAN values of the proximal P3 subchondral zone were significantly lower compared to P1 (*p* = 0.023) but not to P2 (*p* = 0.333). The median EAN values of the proximal P3 trabecular and epiphyseal zone were significantly lower compared to P2 and P1 (*p* = 0.001). Regarding electron density, values of the proximal P3 subchondral, trabecular, and epiphyseal zone were significantly lower compared to P2 (*p* = 0.001–0.044) but not to P1 (*p* = 0.112–0.283). Comparing all proximal zones of P2 with P1, median EAN and ED values did not differ for both machines (*p* = 0.123–1). Regarding the NB, median EAN and ED values of both CT units were significantly lower than all proximal zone values of P1 and P2 (*p* < 0.001–0.025) but not of P3 (*p* = 0.272–1).

*Association of EAN/ED zone values with imaging score, lameness*—Comparing median EAN and ED zone values of normal FJ to FJ with scores of 1 and 2, no significant differences were found for all zones and both CT units (*p* = 0.083–1.0).

If sclerosis or lysis (cleft, cystic lesion) within the subchondral and trabecular ROIs of the sagittal groove of P1 were considered alone (score 2), EAN and ED values also did not significantly differ from normal values (*p* = 0.091–1). Comparison of scores in the PIPJ and DIPJ as well as association of lameness with EAN/ED zone values could not be addressed due to small cell numbers.

## 4. Discussion

The main purpose of the present study was to determine EAN and ED values for normal subchondral and trabecular bone as well as the spongious bone of the NB in the equine distal limb. The presented work was carried out with two different CT units. Both technologies are available at the Vetsuisse Faculty of Zurich and used for clinical examinations as well as for research. Although the calculated descriptive statistics for the EAN and ED bone zones of both units appear very similar and the difference is only in the range of tenth parts, several median EAN zone values were significantly higher, and some ED zones values were significantly lower with SSDECT (Canon) compared to DBSCT (Philips). Differences were even more distinct in the marrow cavity. It appears obvious that the underlying hardware components, i.e., primarily the source-based (Canon) versus detector-based (Philips) approach, as well as the applied vendor-specific algorithms in their software are responsible for the differences. Several studies in human medicine reported inter-vendor variability regarding monochromatic data. Spectral separation, use of filters, cross-scatter between detector layers, and spatial and temporal resolution are some of the factors that differ between DECT technologies [2,30,31,32]. However, there is a lack of standardized parameters with regard to the performance capabilities of DECT in the technical data sheets of the different scanners, which hampers simple comparisons of the available technologies [31]. Therefore, the herein reported significant differences of EAN and ED values between the evaluated DECT technologies appear important and should be considered for clinical and research studies.

With increasing age, EAN zone values in the palmar/plantar aspect of normal FJs increased significantly. The palmar/plantar apical region of MC/T3 and the sesamoid bones are part of the load path in midstance that runs obliquely from the palmar/plantar apical region to the dorsodistal aspect of the epiphysis of MC/T3 [25,33]. Middle-aged to old horses not used for racing were included in the present study. Young horses less than 9 years were excluded, given that, like in children, a higher amount of water in relation to a lower amount of calcium, phosphorus, and collagen and consequently lower EAN values may be expected [26]. Therefore, the age-related increase in EAN most likely represents normal exercise-associated bone adaptation through modelling during life that likely includes an increased bone volume fraction of the subchondral trabecular bone, i.e., an increase of the proportion of heavier atoms, such as calcium [34]. The sagittal groove and the medial and lateral metaphysis of P1 were shown in a model to undergo the highest levels of stress as load increased from stance to walk, trot, and gallop [27], and incomplete proximal fractures of P1 commonly originate from the sagittal groove in middle-aged Warmbloods [28]. Interestingly, in the present study population, EAN values in the subchondral and trabecular zones of normal P1s, measured in a mid-dorsal plane, did not change with age, whereas the proximal P2 subchondral zone exhibited significantly decreasing EAN values, namely a decrease in heavy atoms.

Electron density measurements in all zones of the trabecular and subchondral bone as well as the spongious bone of the NB did not change with age at all. However, in the normal marrow cavity of P1, ED values increased with age and were significantly higher in hind than in front legs. Metabolic demands on the bone marrow may dynamically affect its composition regarding the water, fat, protein, and cellular proportions [35]. Therefore, it appears to be of interest to study EAN and ED values in pathological subchondral and trabecular bone as well as the marrow cavity in the future.

As expected, with both units, EAN and ED values of all trabecular zones in the phalanges were significantly lower than the values of the subchondral zones; median EAN and ED values of the marrow cavity of P1 were the lowest. Lower EAN values may be well explained by the higher proportion of lighter atoms, namely soft tissues, in the trabecular bone and the marrow cavity and a lower proportion of heavier atoms, such as calcium. The lower ED values indicate a lower number of electrons per unit volume of tissue and presumably represent lower cellularity due to more loosely packed cells in the trabecular bone and the marrow cavity [3].

Comparing the normal NB zone and the proximal P1, P2, and P3 zones, it could be clearly seen that the proximal subchondral zone of P3 and the NB zone in the spongious bone (with both CT units) as well as the trabecular zone of P3 (only with DBSCT, Philips) showed significantly lower EAN and ED values, whereas values of the P2 and P1 zones were more alike. This again indicates that the proportion of lighter atoms, i.e., soft tissues, is higher and electron density representing cellularity is lower in the subchondral bone of P3 and the spongious bone of the NB compared to P2 and P1. The elemental compositions of human tissues have been studied intensively; data are used mainly in the field of radiation therapy, radiation protection, and radiobiology [36]. In human skeletal tissues, values were also shown to differ between and within bones [37].

EAN values were significantly higher in the front trabecular zones of P2 and P3 but not in P1 and not in the subchondral zones of all phalanges. The relative magnitude of forces is known to differ between front and hind limbs in the horse, and increased EAN values might represent a partial transfer of the shock absorber function to the trabecular bone in the front limbs [29].

The main purpose of the present study was to determine EAN reference values for normal subchondral bone in joints of the equine distal limb. The study group included horses performing at a low to medium exercise level (leisure, show jumping, dressage). Like other studies [38,39], a high proportion of horses in the present study group had changes in the subchondral bone of the distal metacarpus/metatarsus. Even more had changes in the proximal P1 although they did not display obvious clinical signs, such as lameness or FJ effusion on the included leg. However, a detailed history and results of a lameness examination were lacking in most horses. Therefore, it remains unclear if the findings in the FJ represent subclinical or clinically relevant abnormalities. The ROIs in the FJs of the present study were drawn in the center/middle region of P1 and the condyles of MC/MT3 on a dorsal plane and in the palmar/plantar aspect of the MC/MT condyles on a transverse plane. Most abnormal FJs had a score of 2, i.e., subchondral abnormalities located within a ROI, and only a few FJs exhibited changes outside the ROIs but in the same plane or in more dorsally located areas. Focal subchondral lysis was seen most in P1, i.e., in the sagittal groove and the middle area of the medial fovea. Subchondral sclerosis was also most common in the sagittal groove and the middle and dorsal area of the medial fovea. This is in contrast to a recent study investigating cartilage lesions in the FJ of cadaver limbs of mixed-breed horses. Two regions were identified without any defects, namely the middle area of the lateral and medial fovea of P1, whereas the most frequent regions for cartilage defects were the dorsal aspect of the fovea of P1 (15%), the middle area of the lateral (16%), and the medial part of the condyle of MC/MT3 (18%) [38]. Changes in the subchondral bone were not evaluated in that study; however, one may assume that cartilage lesions precede or are associated with subchondral bone abnormalities in the same location [40]. A recent study used micro-CT in the proximal subchondral bone of P1 in racehorses post-mortem [41]. Their findings demonstrated that differences exist in subchondral bone volumetric bone mineral density and thickness across the proximal osteochondral surface in horses with different training histories. They also found that the subchondral bone of the sagittal groove adapts to race-training in the race-fit groups with an increase in volumetric bone mineral density relative to unraced controls [41]. It was not a major goal of the present study to apply EAN/ED measurements to pathological bone. However, the high number of findings in the fetlock joint was unexpected. Therefore, statistical analysis was performed, and EAN and ED values did not statistically differ between normal zones and scores 1 or 2 or subchondral lysis and sclerosis alone, respectively. Nevertheless, there was a trend for several EAN and ED values to differ between normal and abnormal subchondral bone; therefore, further studies in a larger cohort are needed and appear promising.

This study had several limitations. First, despite a moderate number of investigated legs, the number of included horses was rather low, i.e., multiple legs of the same horse were included. Breeds other than Warmbloods were not investigated, and the association of gender or lameness with EAN values could not be analyzed due to small cell numbers. Second, classification of a normal joint was based on imaging criteria only; information regarding the type and level of exercise, lameness history, and a thorough lameness examination was very limited. Although multiple imaging modalities were used as the gold standard to identify normal bone and joints, the MRI protocol was limited so that articular cartilage defects and soft tissues could not be fully assessed. Nuclear imaging and gross pathology and histology were not performed. Third, measurements of 3–9 ROIs in the central plane of each joint/the NB were acquired and are likely not representative of all areas in the joints/NBs. Placement and size of the ROIs was not adapted to lesions identified in the subchondral bone since evaluation of subchondral bone diseases was not the major goal of the present study. Lastly, an unexpected high number of FJ revealed abnormalities in CT and/or MRI with an unclear clinical significance. However, their number was too small to be addressed statistically.

In summary, further studies in a larger and more standardized cohort should be performed also investigating factors such as breed, type, and level of exercise and results of a clinical and orthopedic examination.

During the past years, CT has become a routine imaging method for the evaluation of bone diseases in the equine distal limb also due to the availability of systems that may be applied to the sedated standing horse. Both DECT systems used in the present study represent projection-based technologies [2,30]. Therefore, a motionless patient is mandatory. Otherwise, the projection data of the high and low kVp scans are not going to be paired, and EAN/ED measurements become unreliable. Consequently, for SSDECT (Canon), recumbent scans under general anesthesia are highly recommended. The latter limitation does not apply to DBSCT (Philips) because this system is always operating in a “dual-energy mode”, and all analyses may be performed retrospectively after CT scanning [2,30]. However, this system is not yet available for the evaluation of a standing horse.

## 5. Conclusions

The present study describes the use of DECT for the quantitative evaluation of bone in horses. The reported EAN and ED values in the subchondral and trabecular bone of the equine distal limb may serve as preliminary reference values. EAN and ED values differed between CT technologies and within and between bones of the distal equine limb. In the future, EAN and ED measurements may be potentially useful to monitor the effect of aging as well as exercise and aid evaluation of subchondral diseases in the horse in clinical and research studies.

## Figures and Tables

**Figure 1 animals-14-01064-f001:**
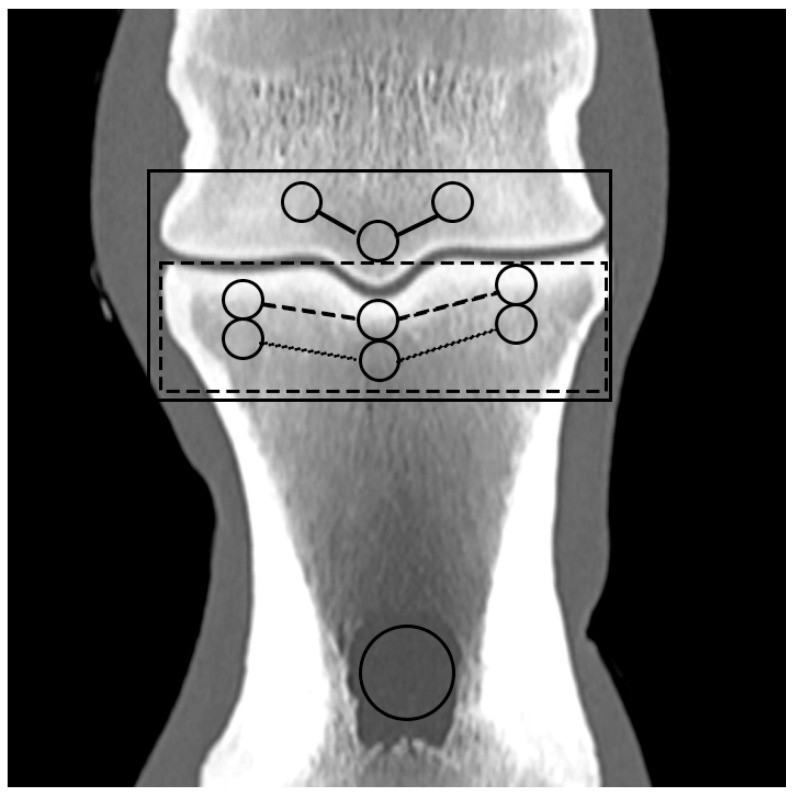
Regions of interest (ROIs; circles) and zones (mean EAN calculated from all ROIs included in an anatomical area; lines) in the dorsal aspect of a front fetlock joint on a mid-dorsal monoenergetic image (detector-based spectral computed tomography, Philips): ROIs in the medial and lateral metacarpal/metatarsal condyle (dorsal MC/MT zone; continuous line), the proximal subchondral and trabecular bone of the proximal phalanx (proximal P1 subchondral/trabecular zone; dashed/dotted line), the marrow cavity of P1, and the proximal epiphyseal P1 zone (dotted rectangular line) and dorsal FJ zone (continuous rectangular line).

**Figure 2 animals-14-01064-f002:**
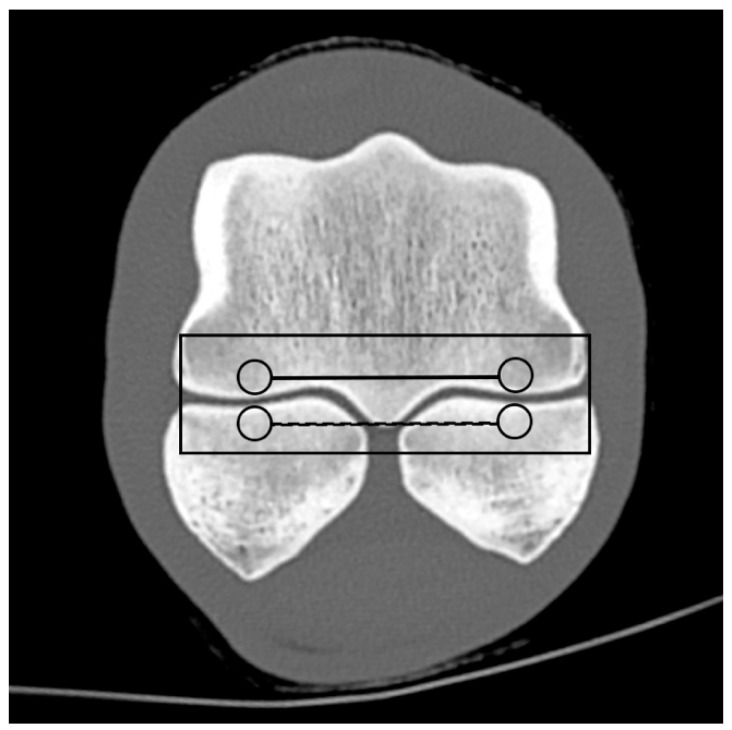
Regions of interest (ROIs; circles) and zones (mean EAN calculated from all ROIs included in an anatomical area; lines) in the palmar/plantar aspect of a front fetlock joint on a transverse monoenergetic image (detector-based spectral computed tomography, Philips) at the mid-level of the proximal sesamoid bones (PSBs): ROIs in the subchondral bone of the medial and lateral metacarpal/metatarsal condyle (palmar/plantar MC/MT zone; continuous line), the dorsal subchondral bone of the medial and lateral PSB (PSB zone; dashed line), and the palmar/plantar FJ zone (continuous rectangular line).

**Figure 3 animals-14-01064-f003:**
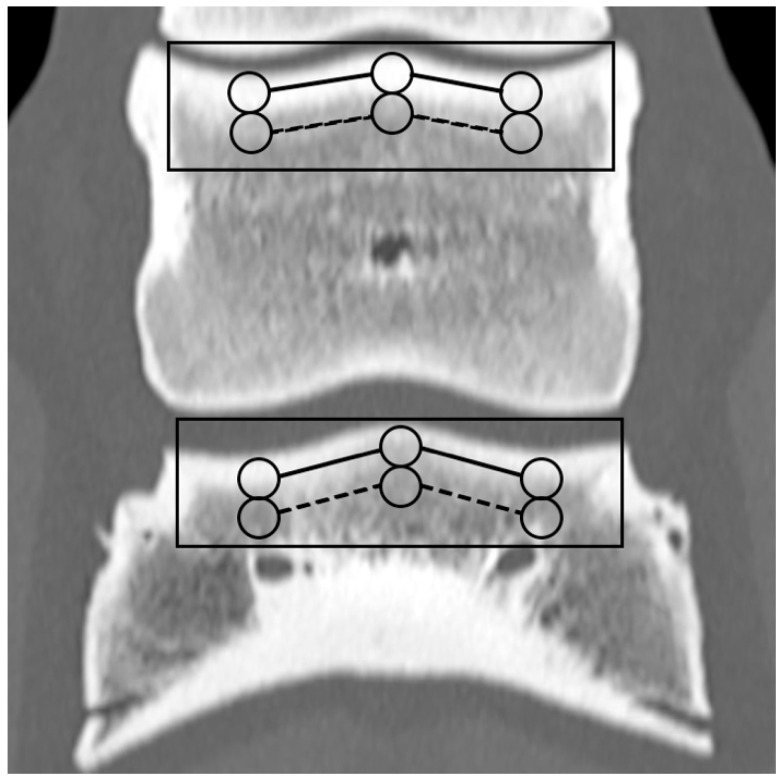
Regions of interest (ROIs; circles) and zones (mean EAN calculated from all ROIs included in an anatomical area; lines) in a front second (P2) and third phalanx (P3) on a mid-dorsal monoenergetic image (detector-based spectral computed tomography, Philips): ROIs in the proximal subchondral and trabecular bone of P2/P3 (proximal P2/P3 subchondral/trabecular zone; continuous/dashed lines) and the proximal epiphyseal P2/3 zone (continuous rectangular line).

**Figure 4 animals-14-01064-f004:**
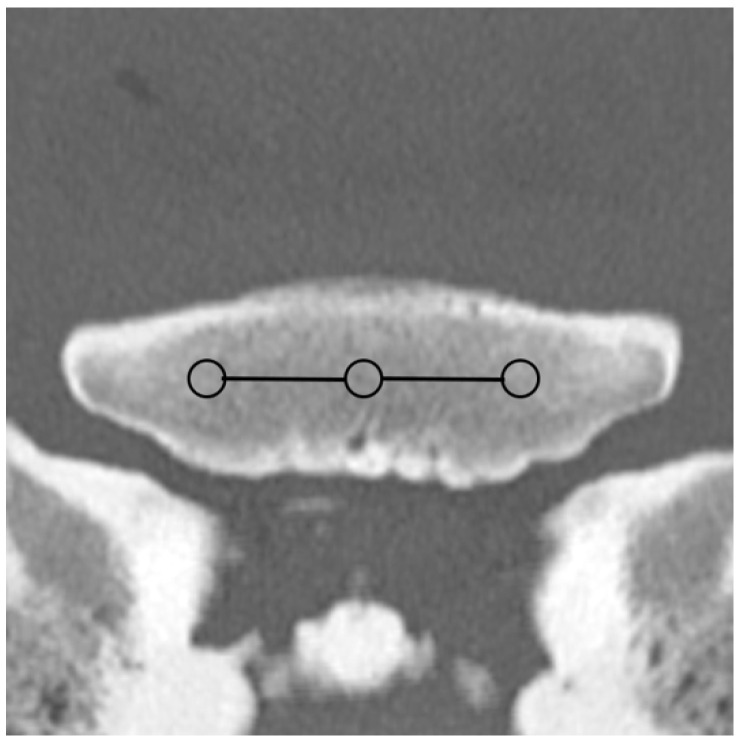
Medial, central, and lateral region of interest (ROI; circles) within the spongious bone of a front navicular bone on a mid-dorsal monoenergetic image (detector-based spectral computed tomography, Philips) and navicular zones (mean EAN calculated from all 3 ROIs; continuous line).

**Table 1 animals-14-01064-t001:** Distribution of scores in the fetlock joint (FJ), the proximal (PIPJ), and distal interphalangeal joint (DIPJ) based on radiographic, conventional computed tomography, and magnetic resonance imaging findings. Score 0: normal joint, score 1: minor findings of no clinical relevance and outside the CT region of interest (ROI), and score 2: major findings of considerable relevance and/or within a CT ROI. DECT: single-source dual energy computed tomography (Canon); DBSCT: detector-based spectral CT (Philips).

	CT Unit	N	FJ	PIPJ	DIPJ
Score 0	DECT	59	10	20	29
	DBSCT	60	10	20	30
Score 1	DECT	21	5	12	4
	DBSCT	21	6	12	3
Score 2	DECT	28	21	4	3
	DBSCT	27	20	4	3

**Table 2 animals-14-01064-t002:** Descriptive statistics of the effective atomic number (EAN) values in 16 zones of normal fetlock joints (FJ), normal proximal and distal interphalangeal joints assessed with single-source dual energy computed tomography (DECT, Canon) and detector-based spectral CT (DBSCT, Philips): metacarpal (MC), metatarsal (MT), proximal sesamoid bone (PSB), first/second/third phalanx (P1/P2/P3), mean (MN), standard deviation (SD), median (MD), standard error (SE), minimum value (MIN), and maximum value (MAX).

Zone	CT Unit	MN	SD	MD	SE	MIN	MAX
Dorsal MC/MT zone	DECT	12.168	0.311	12.185	0.984	11.73	12.53
	DBSCT	11.988	0.330	12.003	0.104	12.01	13.14
Plantar MC/MT zone	DECT	12.600	0.225	12.528	0.711	12.39	13.02
	DBSCT	12.555	0.355	12.620	0.112	12.01	13.14
PSB zone	DECT	12.656	0.999	12.626	0.316	12.56	12.84
	DBSCT	12.532	0.328	12.558	0.104	12.14	12.97
Palmar/plantar FJ zone	DECT	12.628	0.145	12.570	0.046	12.51	12.93
	DBSCT	12.543	0.255	12.491	0.081	12.11	13.00
Proximal P1 subchondral zone	DECT	12.829	0.152	12.835	0.048	12.56	13.00
	DBSCT	12.929	0.205	12.958	0.065	12.58	13.22
Proximal P1 trabecular zone	DECT	11.609	0.237	11.584	0.075	11.11	11.92
	DBSCT	11.355	0.176	11.343	0.555	11.16	11.66
Proximal epiphyseal P1 zone	DECT	12.220	0.161	12.228	0.051	11.60	12.41
	DBSCT	12.142	0.147	12.145	0.047	11.87	12.31
Dorsal FJ zone	DECT	12.202	0.166	12.240	0.525	11.94	12.45
	DBSCT	12.091	0.180	12.097	0.057	11.77	12.32
Proximal P2 subchondral zone	DECT	13.032	0.112	13.045	0.025	12.81	13.21
	DBSCT	13.011	0.175	13.058	0.039	12.61	13.21
Proximal P2 trabecular zone	DECT	11.855	0.215	11.085	0.048	11.52	12.26
	DBSCT	11.549	0.270	11.492	0.060	11.18	12.06
Proximal epiphyseal P2 zone	DECT	12.444	0.115	12.416	0.026	12.22	12.68
	DBSCT	12.280	0.192	12.298	0.043	11.93	12.64
Proximal P3 subchondral zone	DECT	12.505	0.388	12.540	0.072	10.60	12.92
	DBSCT	12.692	0.149	12.707	0.027	12.33	13.03
Proximal P3 trabecular zone	DECT	11.371	0.389	11.32	0.072	10.61	12.5
	DBSCT	10.767	0.293	10.753	0.053	10.07	11.31
Proximal epiphyseal P3 zone	DECT	11.938	0.2	1.94	0.037	11.51	12.35
	DBSCT	11.73	0.162	11.728	0.03	11.29	12.16
Marrow cavity P1	DECT	7.2	0.147	7.2	0.024	6.91	7.44
	DBSCT	5.841	0.166	5.87	0.028	5.22	6.14
Navicular zone	DECT	10.94	0.373	10.9	0.066	10.19	11.71
	DBSCT	10.317	0.332	10.353	0.059	9.51	11

**Table 3 animals-14-01064-t003:** Descriptive statistics of the electron density (ED, 10^23^ e/cm^3^) values in 16 zones of normal fetlock joints (FJ), normal proximal and distal interphalangeal joints assessed with single-source dual energy computed tomography (DECT, Canon) and detector-based spectral CT (DBSCT, Philips): metacarpal (MC), metatarsal (MT), proximal sesamoid bone (PSB), first/second/third phalanx (P1/P2/P3), mean (MN), standard deviation (SD), median (MD), standard error (SE), minimum value (MIN), and maximum value (MAX).

Zone	CT Unit	MN	SD	MD	SE	MIN	MAX
Dorsal MC/MT zone	DECT	4.66	0.18	4.75	0.06	4.41	4.86
	DBSCT	4.64	0.19	4.65	0.06	4.38	4.86
Plantar MC/MT zone	DECT	5.08	0.52	4.97	0.08	4.84	5.58
	DBSCT	5.18	0.26	5.10	0.08	4.86	5.64
PSB zone	DECT	5.20	0.16	5.23	0.05	5.0	5.38
	DBSCT	5.22	0.16	5.22	0.05	4.98	5.48
Palmar/plantar FJ zone	DECT	5.14	0.18	5.12	0.06	4.93	5.75
	DBSCT	5.20	0.17	5.17	0.05	4.99	5.50
Proximal P1 subchondral zone	DECT	5.40	0.12	5.44	0.04	5.25	5.54
	DBSCT	5.49	0.09	5.50	0.03	5.33	5.60
Proximal P1 trabecular zone	DECT	4.30	0.12	4.27	0.04	4.13	4.47
	DBSCT	4.43	0.14	4.42	0.04	4.25	4.61
Proximal epiphyseal P1 zone	DECT	4.85	0.09	4.82	0.03	4.76	5.01
	DBSCT	4.96	0.10	4.96	0.03	4.80	5.10
Dorsal FJ zone	DECT	5.14	0.18	5.12	0.02	4.93	5.47
	DBSCT	4.85	0.10	4.84	0.03	4.71	5.02
Proximal P2 subchondral zone	DECT	5.64	0.07	5.63	0.02	5.47	5.75
	DBSCT	5.67	0.10	5.69	0.02	5.45	5.80
Proximal P2 trabecular zone	DECT	4.35	0.15	4.31	0.03	4.09	4.56
	DBSCT	4.56	0.19	4.55	0.04	4.27	4.94
Proximal epiphyseal P2 zone	DECT	4.99	0.08	4.98	0.02	4.86	5.15
	DBSCT	5.11	0.11	5.10	0.03	4.93	5.35
Proximal P3 subchondral zone	DECT	5.19	0.15	5.13	0.03	5.02	5.59
	DBSCT	5.31	0.14	5.28	0.03	5.04	5.62
Proximal P3 trabecular zone	DECT	4.12	0.16	4.11	0.03	3.79	4.46
	DBSCT	4.27	0.11	4.27	0.02	4.07	4.54
Proximal epiphyseal P3 zone	DECT	4.65	0.12	4.63	0.02	4.44	5.03
	DBSCT	4.79	0.08	4.79	0.01	4.62	4.97
Marrow cavity P1	DECT	3.08	0.20	3.08	0.003	3.04	3.12
	DBSCT	3.11	0.03	3.10	0.006	3.07	3.27
Navicular zone	DECT	3.88	0.13	3.85	0.02	3.71	4.18
	DBSCT	3.89	0.11	3.86	0.02	3.73	4.19

## Data Availability

Data are contained within the article.

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
