# Peer review of "Characterization of Normal Bone in the Equine Distal Limb with Effective Atomic Number and Electron Density Determined with Single-Source Dual Energy and Detector-Based Spectral Computed Tomography"

_animals, 2024, doi:10.3390/ani14071064_

Round 1

Reviewer 1 Report

Comments and Suggestions for Authors

This is a specialized article describing the technical aspects of bone imaging in horses. Given its submission to a themed Special Issue, the manuscript's topic falls within the scope of the journal's subject matter.

Minor comments:

Introduction:

  • Please verify all references. For example, the sentence “Electron density is the measure of probability of an electron being present in a specific location and is expressed as a multiple of 10^23 electrons/mL” cannot be found in the cited reference [6].
  • The sentence “It was hypothesized and then shown in gliomas…” can be removed.
  • Similarly, consider removing or significantly reducing the following paragraph (“Since compounds of different chemical compositions may have similar linear attenuation coefficients…”) as it is not directly related to the study's topic.

Results:

  • Please report all p-values to three decimal places. The range of p-values reported in last paragraphs (like p = 0.12 – 1) is unclear.

Author Response

This is a specialized article describing the technical aspects of bone imaging in horses. Given its submission to a themed Special Issue, the manuscript's topic falls within the scope of the journal's subject matter.

Thank you for your efforts to review our manuscript and for the positive feedback. We appreciate it a lot.

Minor comments:

Introduction:

  • Please verify all references. For example, the sentence “Electron density is the measure of probability of an electron being present in a specific location and is expressed as a multiple of 10^23 electrons/mL” cannot be found in the cited reference [6]. Correct, the references were reordered.
  • The sentence “It was hypothesized and then shown in gliomas…” can be removed. Sentence was removed.
  • Similarly, consider removing or significantly reducing the following paragraph (“Since compounds of different chemical compositions may have similar linear attenuation coefficients…”) as it is not directly related to the study's topic. We shortened the paragraph markedly, however, we kept the cited literature to give an overview of the current research status.

Results:

  • Please report all p-values to three decimal places. Three decimal places were inserted wherever applicable. The range of p-values reported in last paragraphs (like p = 0.12 – 1) is unclear. The range refers to all non-significant variables. Listing all p-values would be confusing and elongate the results section, therefore we decided to work with ranges.

Reviewer 2 Report

Comments and Suggestions for Authors

Interesting study, a well-written paper.

There are a few typographical errors to correct.  Section 2.4, paragraph 2, "A single ROI was also applied to the marrow cavity of P1 (Figure 3)."  I believe this should reference Figure 1.

Section 3 (Results) Imaging score "were by far less common in the PGB (8% and 12.5%) than"  I believe this should be PSB not PGB.

Tables 2 and 3 description "metacarpal (MC), metatarsal (MT), proximal suspensory bone (PSB), first/second/third" I believe this should be proximal sesamoid bone, not proximal suspensory bone.

Author Response

Interesting study, a well-written paper.

Thank you for your efforts to review our manuscript and for the positive feedback. We appreciate it a lot.

There are a few typographical errors to correct.  Section 2.4, paragraph 2, "A single ROI was also applied to the marrow cavity of P1 (Figure 3)."  I believe this should reference Figure 1. Correct, this was changed.

Section 3 (Results) Imaging score "were by far less common in the PGB (8% and 12.5%) than"  I believe this should be PSB not PGB. Correct, this was changed.

Tables 2 and 3 description "metacarpal (MC), metatarsal (MT), proximal suspensory bone (PSB), first/second/third" I believe this should be proximal sesamoid bone, not proximal suspensory bone. Correct, this was changed.

Reviewer 3 Report

Comments and Suggestions for Authors

The authors are aware of the limitations of the study and report them with due care. Despite this, the research is very interesting, addresses a very current topic (early diagnosis) and, above all, highlights some important points for the evolution and application of this diagnostic technique in horses. In the near future, in fact, EAN and ED measurements could potentially be useful for monitoring the effect of physical exercise and the clinical evaluation of subchondral diseases.

Author Response

The authors are aware of the limitations of the study and report them with due care. Despite this, the research is very interesting, addresses a very current topic (early diagnosis) and, above all, highlights some important points for the evolution and application of this diagnostic technique in horses. In the near future, in fact, EAN and ED measurements could potentially be useful for monitoring the effect of physical exercise and the clinical evaluation of subchondral diseases.

Thank you for your efforts to review our manuscript and for the positive feedback. We appreciate it a lot.